# X-ray Photoemission Spectroscopy Study of Uniaxial Magnetic Anisotropy Induced in a Ni Layer Deposited on a LiNbO_3_ Substrate

**DOI:** 10.3390/nano11041024

**Published:** 2021-04-16

**Authors:** Akinobu Yamaguchi, Takuo Ohkochi, Masaki Oura, Keisuke Yamada, Tsunemasa Saiki, Satoru Suzuki, Yuichi Utsumi, Aiko Nakao

**Affiliations:** 1Laboratory of Advanced Science and Technology for Industry, University of Hyogo, 3-1-2 Kouto, Kamigori, Hyogo 678-1205, Japan; ssuzuki@lasti.u-hyogo.ac.jp (S.S.); utsumi@lasti.u-hyogo.ac.jp (Y.U.); a.nakao@aoni.waseda.jp (A.N.); 2RIKEN SPring-8 Center, 1-1-1 Kouto, Sayo, Hyogo 679-5148, Japan; o-taku@spring8.or.jp (T.O.); oura@spring8.or.jp (M.O.); 3Japan Synchrotron Radiation Research Institute, 1-1-1 Kouto, Sayo, Hyogo 679-5198, Japan; 4Department of Chemistry and Biomolecular Science, Faculty of Engineering, Gifu University, 1-1 Yanagido, Gifu City, Gifu 501-1193, Japan; yamada_k@gifu-u.ac.jp; 5Manufacturing Technology Department, Hyogo Prefectural Institute of Technology, 3-1-12 Yukihira, Suma, Kobe 654-0037, Japan; saiki@hyogo-kg.jp; 6Research Organization for Nano and Life Innovation, Waseda University, 513 Tsurumaki-cho, Waseda, Shijuku-ku, Tokyo 162-0041, Japan

**Keywords:** heterojunction, magnetic anisotropy, magnetoelastic effect, X-ray photoemission spectroscopy, X-ray magnetic circular dichroism photoemission electron microscopy

## Abstract

The competition between magnetic shape anisotropy and the induced uniaxial magnetic anisotropy in the heterojunction between a ferromagnetic layer and a ferroelectric substrate serves to control magnetic domain structures as well as magnetization reversal characteristics. The uniaxial magnetic anisotropy, originating from the symmetry breaking effect in the heterojunction, plays a significant role in modifying the characteristics of magnetization dynamics. Magnetoelastic phenomena are known to generate uniaxial magnetic anisotropy; however, the interfacial electronic states that may contribute to the uniaxial magnetic anisotropy have not yet been adequately investigated. Here, we report experimental evidence concerning the binding energy change in the ferromagnetic layer/ferroelectric substrate heterojunction using X-ray photoemission spectroscopy. The binding energy shifts, corresponding to the chemical shifts, reveal the binding states near the interface. Our results shed light on the origin of the uniaxial magnetic anisotropy induced from the heterojunction. This knowledge can provide a means for the simultaneous control of magnetism, mechanics, and electronics in a nano/microsystem consisting of ferromagnetic/ferroelectric materials.

## 1. Introduction

The heterojunction, or heterointerface, is of great importance in a wide range of applications, such as thermal-barrier coatings, nanocomposites, electronic and optical devices, giant magnetoresistive and magnetic tunnel junctions, various catalysts, and batteries. In these various applications, for example, in field-effect transistors, solar cells, and light emitting devices, the band alignments of the heterojunction are crucial factors. Novel multiferroic materials [1] might also be created by heterojunctions. The characteristics of a heterojunction vary greatly depending on the surface termination species [2,3,4,5,6,7,8,9,10,11,12,13,14,15,16,17,18,19,20]. As pointed out by Stöhr [8], “symmetry breaking and boning at interfaces leads to a variety of anisotropy phenomena in transition metal sandwiches and multilayers”. For example, in the Ni/Cu(100), Fe/Cu(100), and Fe/MgO systems, the magnetic anisotropy is induced by the orbital-coupling and spin-flip at the interface [3,4,17]. Candidate phenomena that are useful for inducing magnetic anisotropy are the Dzyaloshinskii-Moriya interaction (DMI) [21,22], Rashba interaction [23], magnetoelastic effect, and magnetostriction [24,25]. These interactions and effects can also be induced by symmetry breaking and bonding at the interface.

In general, desired magnetic domain structures can be formed by fabricating the films into nano/microstructures, in other words, by properly tailoring the magnetic shape anisotropy [26,27,28,29,30]. In artificial multilayer systems, exchange bias or synthetic antiferromagnetic coupling, which is derived from the quantum interference generated via the heterojunction between ferromagnetic and antiferromagnetic layers, is generally used to control the magnetic properties [7,8,10,11,12,13,14,15,16,17,18]. As for the uniaxial magnetic anisotropy that arises from the heterojunction, some investigations have been reported in the systems consisting of a thin ferromagnetic layer and a ferroelectric substrate, e.g., lithium niobate (LiNbO_3_) [31,32,33,34,35,36,37,38,39]. Several of our previous studies have examined unusual magnetic characteristics found in a 30 nm thick Ni wire fabricated on a single crystal Y-cut 128° LiNbO_3_ substrate [33,34,35,36]. Figure 1a shows a photoemission electron microscopy (PEEM) topographical image obtained by an ultraviolet lamp, (Figure 1b,c) X-ray magnetic circular dichroism photoemission electron microscopy (XMCD-PEEM) images at the Ni *L*_3_-edge, and (Figure 1d) a pair of dichroic X-ray absorption spectra around the Ni *L*_2,3_-edge of microfabricated 30 nm thick Ni patterns on a LiNbO_3_ substrate, obtained at the soft X-ray beam line BL17SU at SPring-8 [39]. The white and black contrast in Figure 1b,c denote the magnetization aligning parallel and antiparallel to the X-ray direction, respectively. We found that uniaxial magnetic anisotropy was induced in the Ni layer on the LiNbO_3_ substrate due to the formation of the heterojunction; the uniaxial magnetic anisotropy forced the Ni magnetizations to stochastically align parallel or antiparallel to the *X*-axis of the LiNbO_3_ substrate (indicated by an arrow in region (i) of Figure 1b). As previously reported [33,34,35,36,37,38], a stripe domain structure is naturally formed when the wire is aligned perpendicular to the *X*-axis of the LiNbO_3_ substrate, even in the absence of an external magnetic field, due to the competition between magnetic shape anisotropy and uniaxial magnetic anisotropy induced by the heterojunction (region (ii) of Figure 1b) [33,34,35,36,37,38]. The magnetic domain width is dependent on the width of wire, being almost equal to the width of the wire. The XMCD spectrum of the Ni layer on the LiNbO_3_ substrate in Figure 1d was obtained by taking the difference between the X-ray absorption spectra (XAS) spectra extracted from the white (magnetization parallel to X-ray) and black (antiparallel) regions. These XAS spectra revealed that the Ni layer was not being clearly oxidized. The shape to the XMCD spectrum obtained here was almost the same as that of conventional Ni plain films [4,9]. The drastic change in the magnetoresistance effect of the Ni wires elongated perpendicular and parallel to the *X*-axis of the LiNbO_3_ substrate, as shown in Figure 4 of Ref. [36], reflected the unusual domain structures, as observed in Figure 1.

What induces uniaxial magnetic anisotropy? To investigate the physical origin of uniaxial magnetic anisotropy generated in the Ni layer fabricated on the LiNbO_3_ substrate, we performed depth-dependent X-ray photoelectron spectroscopy (XPS).

## 2. Sample Fabrication and Methods

Using a magnetron sputtering machine, we deposited several specimens that were composed of a 30 nm thick Ni layer on a Y-cut 128° LiNbO_3_ substrate, or a Si substrate with a naturally oxidized layer formed at the surface, at room temperature (SiO_2(nat)_/Si). The prepared three systems were as follows #1: Au-cap (4 nm)/Ni (30 nm)/LiNbO_3_, #2: Au-cap (4 nm)/Ni (12.5 nm)/Au (12 nm)/LiNbO_3_, and #3: Au-cap (4 nm)/Ni (12.5 nm)/SiO_2(nat)_/Si. In addition, as reference samples, two systems of #4: Au-cap(4 nm)/Ni_81_Fe_19_(12.5 nm)/LiNbO_3_ and #5: Au-cap(4 nm)/Ni_81_Fe_19_(12.5 nm)/SiO_2(nat)_/Si were prepared. The XPS measurements were performed using a VG ESCALAB 250 spectrometer (Thermo Fisher Scientific K.K. Tokyo, Japan), employing monochromatic Al Kα X-ray radiation (1486.6 eV) at room temperature. The system was operated at 200 W, and its acceleration voltage was set to be 15 kV. The base pressure of the analysis chamber was less than 10^−8^ Pa. A depth profile was measured using an Ar ion sputtering gun with a beam voltage of 3 keV. The etching rate was approximately 3 nm/min = 0.05 nm/s for Si. Even though the etching rate depends on the materials and photoelectron mean free path on the binding energies of the core transitions, we confirmed through a rough estimation that a typical full width of half maximum of a differential of the slope of the XPS intensity (~40 s, ~20 Å) was consistent with a convolution of the mean free path (10 ± 2–3 Å for the electron kinetic energy of 600–1400 eV) and the interface roughness of the layered samples (10–20 Å) [40]. The XPS film thickness dependence (XPS depth profile) was repeatedly measured every time after the Ar sputtering for ~10 s (the sputtering was paused during the XPS measurements). When the milled surface reached an insulating substrate, a charge balancing treatment was needed. In order to compensate for the surface charge, we used a low-energy electron flood gun. During the course of the electron dozing, a charge neutralizer was referred to in order to over/undercompensate the charge. To further enhance the charge compensation, an earthed conductive grid and a carbon conductive tape were attached around the affected area. In addition, we rechecked the obtained XPS spectra repeatedly to verify reproducibility, including the check of the Au 4*f* and Ni 2*p* peak positions relative to the reference values.

## 3. Results and Discussion

Figure 2a shows the intensity profiles of Au 4*f*, Ni 2*p*, Nd 3*d*, and O 1*s* core-level peaks as a function of Ar ion etching time for system #1. First, the Au 4*f* peak derived from the Au capping layer appeared for approximately 100 s, which gradually disappeared as the Ni 2*p* peak increased, that is, as the Ni layer underneath the Au cap was exposed. When approaching the Ni/LiNbO_3_ interface (near 1000 s), the O 1*s* and Nb 3*d* signals appeared while the Ni 2*p* peak gradually decreased. This etching profile confirmed the correct preparation of the desired system. Next, we focused on the depth dependence of the XPS spectrum shape for each element. Here, let us note that the arrangement of each spectrum was set in order to clarify the temporal transition of the spectral shape. In addition, to track the etching time dependence of the spectrum, some typical etching times are described for the representative spectrum. First, Figure 2b shows the XPS spectra of Nb 3*d* as a function of etching time. According to Refs. [41,42], XPS peaks from Nb 3*d* are decomposed into 3*d* 3/2 and 3*d* 5/2 contributions: LiNbO_3_ (209.41 and 206.63 eV), Nb_2_O_5_ (209.78 eV, 207.03 eV), NbO_2_ (208.23 and 205.67 eV), and NbO (207.47 and 204.67 eV). We found that the subpeak at the binding energy (*E_B_*) of about 204 eV increased with increasing etching time after 1050 s. To understand this behavior and to clarify the mechanism, the fittings were performed in the XPS spectra obtained at 1050 s and 1300 s, as shown in Figure 2c,d, respectively. As a result, the XPS spectrum at 1050 s was well-correlated to the fitting calculated by the assumption that these peaks were only the LiNbO_3_ substrate. The XPS spectrum at 1300 s was in good agreement with the fitting obtained from the LiNbO_3_, Nb_2_O_5_, and NbO. In particular, abrupt increase in the XPS peak at about 204 eV was considered to be derived from NbO. Consequently, the following result was obtained: NbO was seen just after the pure Ni layer was removed and the LiNbO_3_ substrate started to appear (~930 s). Then, until the Ni/LiNbO_3_ intermixed layer was completely removed (~1300 s), the XPS intensity of Ni 2*p* decreased with increasing etching time, as shown in Figure 2e. The LiNbO_3_ layer partially transformed to NbO and Nb_2_O_5_-like phases, which were devoid of lithium because the Li was preferentially etched by the Ar+ sputtering [42]. Accompanying the generation of the NbO and Nb_2_O_5_ XPS peak, the O 1*s* XPS peak also shifted to a lower binding energy, as shown in Figure 2f. Conversely, as shown in Figure 2e, the Ni 2*p* peak shifted to a higher binding energy as the interface was approached. Note that an XPS peak derived from NiO was not observed for all depth ranges; we can safely assume that the surface Ni was perfectly protected against oxidization by the Au cap, and that Ni hardly reacted with the LiNbO_3_ substrate.

To confirm the Ni film thickness dependence of the XPS spectra, the same measurement was performed for a sample with the same multilayer configuration as #1, but the Ni thickness was changed to 12.5 nm. The overall trend in the XPS spectra was practically the same as that shown in Figure 2a–f (not shown here), in that the XPS peak of Ni 2*p* always shifted to a higher binding energy near the interface with the LiNbO_3_ substrate. This result suggests that the electron state can only be modulated around the interface, as long as the Ni thickness is at least thicker than 12.5 nm.

Next, in a similar procedure, we measured the XPS spectra of system #2: the Au-cap/Ni/Au/LiNbO_3_ substrate, to investigate the interface effect. The intensity profiles of the XPS peaks, together with the variation in the shapes of the Au 4*f*, Ni 2*p*, and O 1*s* spectra with respect to the etching time, are shown in Figure 3a–d. No shift in the Au 4*f* XPS spectra was observed for both sides of the interlayer with the Ni layer and the LiNbO_3_ substrate (Figure 3b), while the Ni 2*p* XPS peak shifted to a lower binding energy as the Ni approached the Au layer (Figure 3c). The trend in the Ni 2*p* peak shift was opposite to the case of the system consisting of the Au-cap/Ni/LiNbO_3_ substrate, as shown in Figure 2e. This result indicates that the combination of materials at the interface plays a significant role in the modulation of the bonding state of the interfacial Ni.

To investigate the substrate dependence of the XPS spectra, we measured the XPS of a system consisting of an Au capping layer and a Ni layer deposited on a slightly oxidized Si substrate (#3). The etching profile is shown in Figure 4a. The XPS spectra of Ni 2*p* are displayed in Figure 4b. As shown in Figure 4b, we found that the Ni 2*p* XPS spectra abruptly changed between 340 s and 380 s, near the interface. The newly appeared peak at approximately 855 eV, being a higher binding energy than the Ni 2*p* fundamental peak position, corresponded to the peak derived from Ni oxide (NiO). When approaching the interface and surface of Si substrate, Ni 2*p* metallic peak almost disappeared after 420 s. Next, we checked the XPS spectra of O 1*s* and Si 1*s*. Figure 4c,d indicates that the XPS peak positions of O 1*s* and Si 1*s* also shifted, synchronizing with the NiO forming process. This result indicates that the oxygen at the Ni/SiO interface preferentially bonds with nickel, which results in the formation of NiO, even if the oxidized Si layer is vanishingly thin (unlike the case of thermally oxidized thick SiO_2_). NiO is a well-known antiferromagnetic material. These results indicate that the NiO formed at the interface might act as a pinning layer against the magnetic domain wall displacement, or the magnetization reversal.

Next, to compare the XPS spectra of Ni_81_Fe_19_ layers on LiNbO_3_ and Si substrates with that of Ni layers on these substrates, we measured the XPS spectra of systems #4 and #5 in a similar way. As seen from the intensity profiles in Figure 5a, the spectrum of each element around 400 s would be useful to closely investigate the Ni_81_Fe_19_/LiNbO_3_ interface states. The shape of the Fe 2*p* spectra shown in Figure 5b remained unchanged through all the depth range, although the overall peak intensity decreased as the Ni_81_Fe_19_ layer was etched out. The XPS peak positions of Ni 2*p* (Figure 5c) and O 1*s* (Figure 5d) slightly depended on the etching time. In the O 1*s*, the peak position slightly shifted toward binding energy during 330–420 s, and then settled to *E_B_*~531 eV. The Ni 2*p* peak appeared to shift to the higher binding energy when approaching the substrate; however, it was subtler than the case of Ni/LiNbO_3_ (Figure 2c) and Ni/Au interfaces (Figure 3c). These results indicate that the bonding state of Ni_81_Fe_19_ layer is not strongly influenced by the LiNbO_3_ substrate.

Figure 6 shows similar plots to Figure 2, Figure 3, Figure 4 and Figure 5 for the systems #5: Ni_81_Fe_19_ deposited on a naturally oxidized Si substrate. Ni 2*p* spectrum (Figure 6b) shows a single peak indicative of metallic Ni (*E*_B_ = 853.5 eV) until the surface reached Ni/SiO*_x_* interface (<350 s) but the subpeak that derived from NiO (*E*_B_ = 854.7 eV) became dominant in the middle of the interface (>420 s). The XPS spectra of Si tended to drastically change during 330 – 390 s, as shown in Figure 6d. The O 1*s* spectrum (Figure 6c) showed a tiny peak at ~530 eV before reaching the interface (330 s), then it grew and settled to the fixed position of 532.5 eV, where we could not distinguish the attribution (NiO or SiO_2_), but presumably the oxygen was almost fully transferred to the Ni (or Fe) site, considering that the peak attributable to SiO_2_ (*E*_B_ = 103–104 eV) was not existent through all the temporal range in the Si 1*s* spectrum (Figure 6d).

Therefore, it appears that the binding strength of Ni with respect to O is large and hence the formation of a NiO layer is inevitable at the interface between the Ni layer and the (naturally) oxidized Si substrate. This is important when investigating the pinning site against the domain wall displacement in a system consisting of a Ni film and a SiO_2_ (or a Si) substrate. In contrast, as is obvious from the cases of the systems #1 and #3, the Ni layer is not oxidized at the interface between Ni and the LiNbO_3_ substrate. In other words, the oxygen in the LiNbO_3_ is very stable and rarely causes chemical reaction with contacting layers such as Ni. Therefore, LiNbO_3_ does not lose its piezoelectric feature, which in turn conveys strong uniaxial magnetic anisotropy to the adjacent magnetic layer.

Here, we discuss the systematic peak shifts in the Ni 2*p* core photoemission that occurred in samples #1 and #2, that is, the Ni 2*p* peak shifted to a lower binding energy when approaching the interface of Ni/Au interface, whereas it went to a higher binding energy in the case of Ni/LiNbO_3_ interface. Because the Ni, which is in contact with an oxide substrate, tends to firmly attach with each other by a Coulomb attractive force if not chemically reacted to the point of forming NiO, the outer shell electrons of the Ni are slightly attracted to the oxygen site, which leads to the increase in the binding energy of the core electrons. In the case of the Ni in the vicinity of a noble metal like Au, by contrast, the Ni on the Au side form a dangling-bond and hence the binding energy of the 2*p* core tends to be weakened. Summarizing the above speculations, the Ni and the LiNbO_3_ layers are adhesive to each other by indirect Ni–O Coulomb interaction, while they do not chemically react with each other and the LiNbO_3_ maintains the bulk property (piezoelectric material) even at the surface/interface. Therefore, the strong uniaxial anisotropy at the Ni/LiNbO_3_ interface may be induced because the magnetoelastic effect due to the crystallographic striction of the ferroelectric LiNbO_3_ is conveyed to the Ni layer quite effectively.

Therefore, the heterojunction between the magnetic metallic film and the ferroelectric substrate induces the uniaxial magnetic anisotropy via the interface effect including the magnetoelastic effect, magnetostriction [5,24,25], DMI [21,22], and electron wave function interference at the interface through the Rashba effect [23]. This investigation revealed an electronic state modulated by the heterojunction. This modulation of the electronic state via the heterojunction enables the generation of magnetic anisotropy. As a result, the combination of the shape magnetic anisotropy and uniaxial anisotropy induced by the heterojunction enables the magnetic domain structure formation in microscale Ni wires fabricated on a LiNbO_3_ substrate to be controlled. The voltage application of the system consisting of ferroelectric and ferromagnetic materials might enable us to control the magnetization reversal and magnetic anisotropy [31,32,37,38].

## 4. Conclusions

We conducted XPS analyses to unveil the mechanism of the uniaxial magnetic anisotropy generation in Ni deposited on LiNbO_3_ substrates. Fine structures of the XPS spectra vary depending on the combination of the interface materials. The XPS results revealed the biding energy modulation of Ni layer on LiNbO_3,_ and the formation of NiO layer on naturally oxidized Si and SiO_2_/Si substrates. These changes in the XPS spectra indicated that the unusual uniaxial magnetic anisotropy generated in the Ni/LiNbO_3_ system effectively induced by the internal stress and band modulation associated with the heterojunction formation. We found that the Ni/LiNbO_3_ heterojunction was fairly firm but never formed any weird interface materials by chemical reactions without annealing. In addition, a NiO layer always formed at the interface between Ni and SiO_2_. These results indicate that the symmetry of the layer structure and, in turn, the magnetic anisotropy of the film are strongly linked to the ferroelectric substrate, and chemical reactions are unavoidable near the interface depending on the composition of the films. These chemical couplings near the interface between ferromagnetic and ferroelectric layers are critically important to understanding the physical and chemical mechanisms causing functional properties. From the viewpoint of the material design, our study provides valuable information. The formation of a heterojunction may be a promising technique to develop novel functional materials.

## Figures and Tables

**Figure 1 nanomaterials-11-01024-f001:**
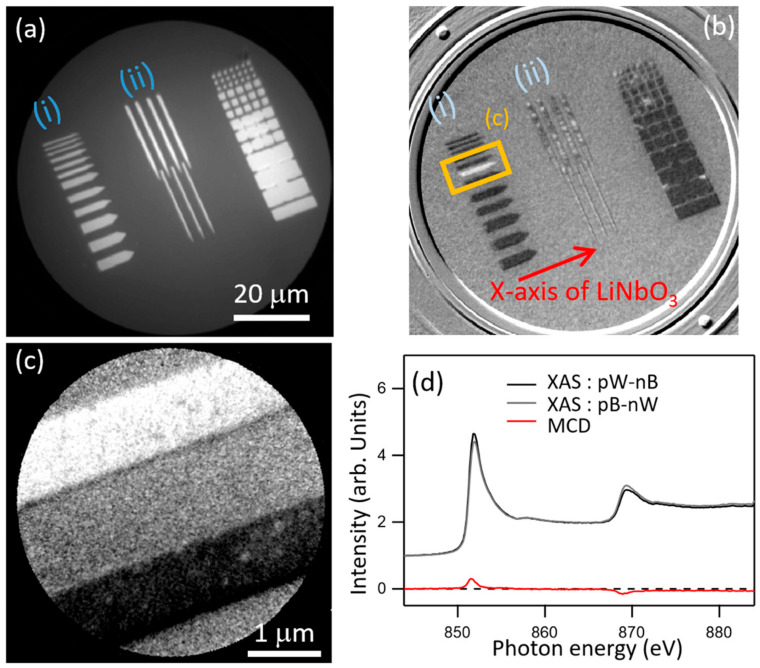
(**a**) Photoemission electron microscopy (PEEM) topographical image and (**b**) X-ray magnetic circular dichroism (XMCD)-PEEM image of 30 nm thick Ni dots and wires deposited on a Y-cut 128° LiNbO_3_ substrate. These dots and wires aligned (i) parallel and (ii) perpendicular to the *X*-axis of the LiNbO_3_ substrate. (**c**) Magnified XMCD-PEEM images of Ni wires, whose magnetizations are parallel and antiparallel to the synchrotron radiation light. (**d**) X-ray absorption spectra (XAS) obtained at the helicity of right (p) and left (n) for the white (W) and black (B) colored contrasts. XAS: pW-nB and XAS: pB-nW are defined as the XAS spectra of intensity (pW)/(nB) and (pB)/(nW), respectively. XMCD spectra of Ni wire calculated from the subtraction of [XAS: pW-nB] − [XAS: pB-nB].

**Figure 2 nanomaterials-11-01024-f002:**
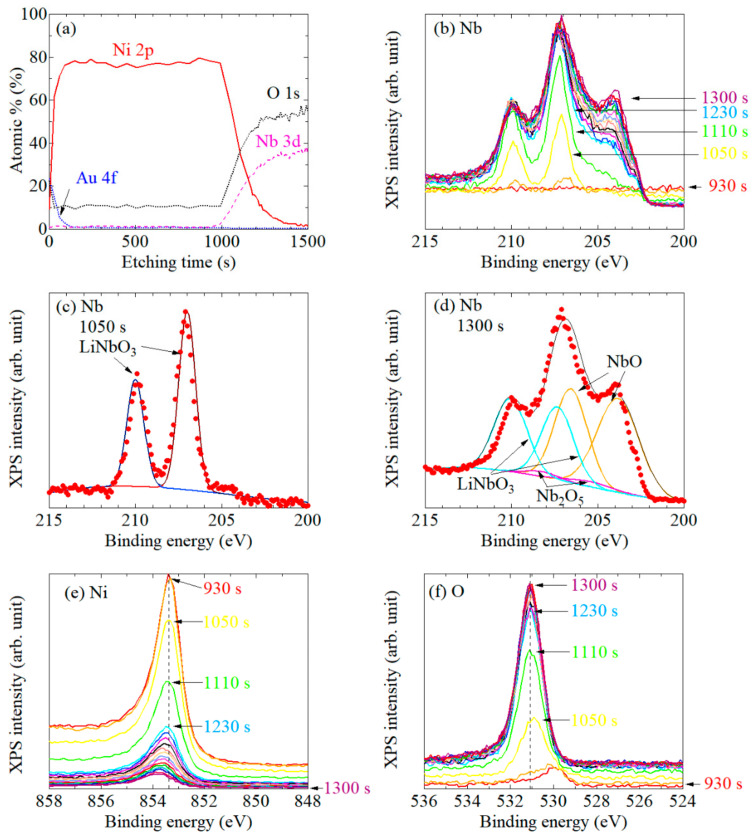
(**a**) Ar ion etching profiles for the system #1: the Au-cap/Ni/LiNbO_3_ substrate. XPS spectra of (**b**–**d**) Nb 3*d*, (**e**) Ni 2*p*, and (**f**) O 1*s*. The numbers correspond to the etching times in the etching profile in panel (**a**). Nd 3*d* peaks in the XPS spectra at (**c**) 1050 s and (**d**) 1300 s are fitted. For clarity, the spectra are vertically shifted. The arrangement of each spectrum is set to the forward direction or the reverse direction for each etching time. Hereinafter, in all the figures, a similar drawing is used.

**Figure 3 nanomaterials-11-01024-f003:**
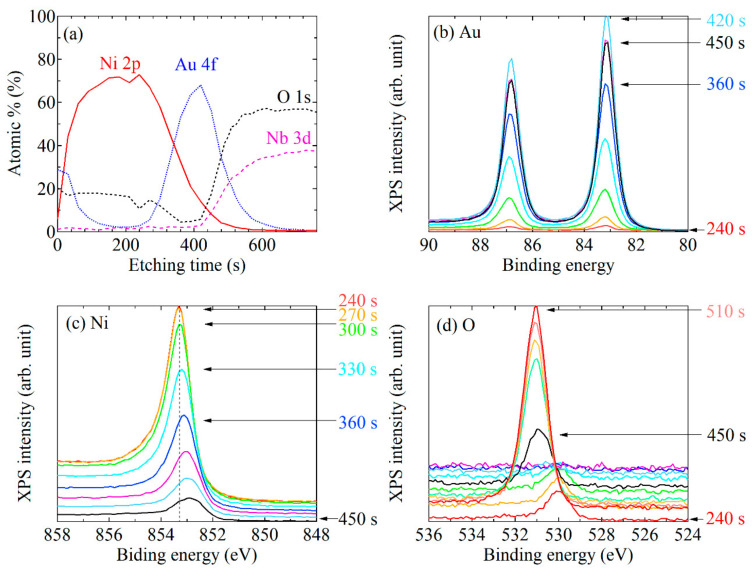
(**a**) Ar ion etching profiles for the system #2: the Au-cap/Ni/Au/LiNbO_3_ substrate. XPS spectra of (**b**) Au 4*f*, (**c**) Ni 2*p*, and (**d**) O 1*s*.

**Figure 4 nanomaterials-11-01024-f004:**
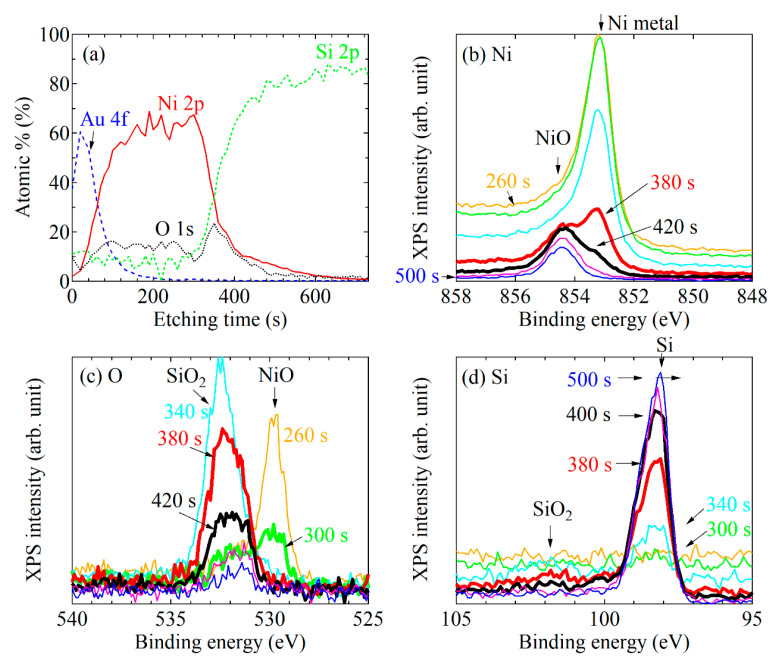
(**a**) Ar ion etching profiles of the system #3: the Au-cap/Ni/naturally oxidized SiO_2_/Si substrate. XPS spectra of (**b**) Ni 2*p*, (**c**) O 1*s*, and (**d**) Si 2*p*.

**Figure 5 nanomaterials-11-01024-f005:**
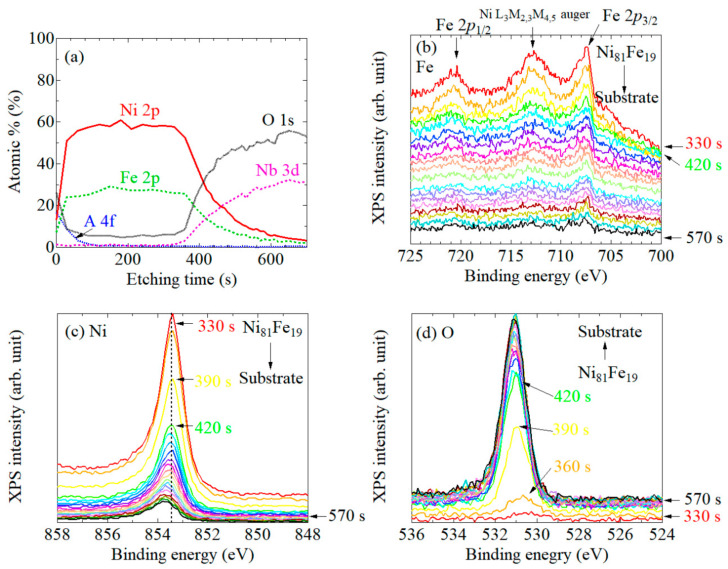
(**a**) Ar ion etching profiles of the system #4: the Au-cap/Ni_81_Fe_19_/LiNbO_3_ substrate. XPS spectra of (**b**) Fe 2*p*, (**c**) Ni 2*p*, and (**d**) O 1*s*.

**Figure 6 nanomaterials-11-01024-f006:**
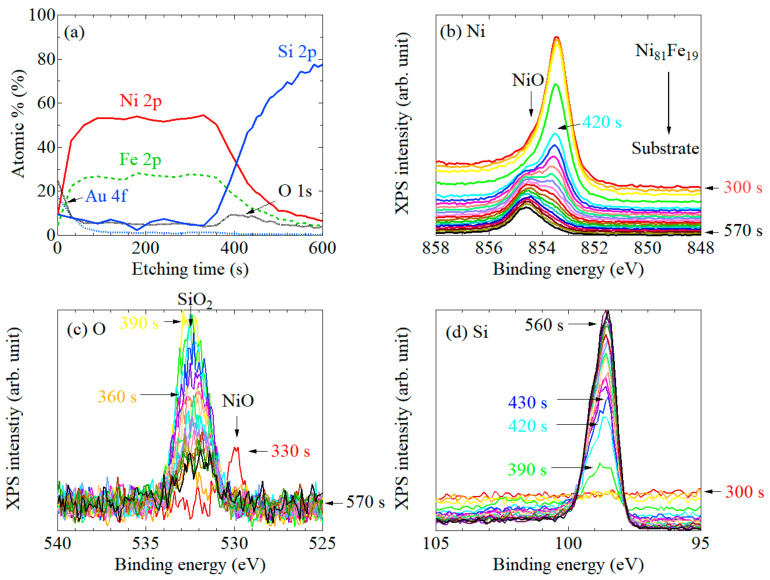
(**a**) Ar ion etching profiles of the system #5: the Au-cap/Ni_81_Fe_19_/naturally oxidized SiO_2_/Si substrate. XPS spectra of (**b**) Ni 2*p*, (**c**) O 1*s*, and (**d**) Si 2*p*.

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
