# Peer review of "X-ray Photoemission Spectroscopy Study of Uniaxial Magnetic Anisotropy Induced in a Ni Layer Deposited on a LiNbO_3_ Substrate"

_nanomaterials, 2021, doi:10.3390/nano11041024_

Round 1
Reviewer 1 Report
The authors report experimental evidence concerning the binding energy change in the ferro-magnetic layer/ferroelectric substrate heterojunction by using X-ray photoemission spectroscopy.
I have several minor comments
- Can you give the magnetic domains size of the LiNbO3?
- Please state why in the parallel to the X-axis of LiNbO3 substrate, there is only one wire that is white (parallel to X-ray)?
Typos I found:
- Line 40: Nnovel – Novel
- Line 62: a ultraviolet - an ultraviolet
- Line 68: due the - due to the
- Line 78: XAS - X- ray absorption spectra (XAS)
- Line 78: write – white
- Line 140: 240 eV -204 eV
Author Response
Response to the Reviewer 1
Thank you very much for your instructive and helpful examination on our work. We are very delighted to receive the Reviewer’s criticisms. Following the Reviewer’s criticisms and comments, we have revised our manuscript for publication in Nanomaterials. Below, we address our replies to all the issues raised in our revised manuscript:
[Reviewer’s criticisms (1)]
The authors report experimental evidence concerning the binding energy change in the ferro-magnetic layer/ferroelectric substrate heterojunction by using X-ray photoemission spectroscopy.
I have several minor comments
1) Can you give the magnetic domains size of the LiNbO3?
2) Please state why in the parallel to the X-axis of LiNbO3 substrate, there is only one wire that is white (parallel to X-ray)?
[Our response (1)]
We appreciate the Reviewer’s criticisms and instructive advice. The magnetic domain size depends on the Ni wire width, deposited on the LiNbO3 substrate. The domain width is almost equal to the wire width. Magnetization in single domain in region (i) of Fig. 1(b) is stochastically aligned parallel or antiparallel to the X-axis of the LiNbO3 substrate. Following the Reviewer’s criticisms, we have revised the sentences from line 17 to line 26 in the 2nd paragraph of Introduction. (line 67 to line 75)
[Reviewer’s criticisms (2)]
Typos I found:
Line 40: Nnovel – Novel
Line 62: a ultraviolet - an ultraviolet
Line 68: due the - due to the
Line 78: XAS - X- ray absorption spectra (XAS)
Line 78: write – white
Line 140: 240 eV -204 eV
[Our response (2)]
We appreciate the Reviewer’s helpful and instructive founds. Thanks to the Reviewer, we have corrected them.
We thank the Reviewer for his/her helpful comments. As the Reviewer pointed out, we have revised our manuscript with clearer discussion. We hope you will find our responses and revision to be appropriate.
Yours sincerely,
Akinobu Yamaguchi
---------------------------------------------------------------------
Akinobu Yamaguchi (Ph.D.)
Laboratory of Advanced Science and Technology for Industry, University of Hyogo 3-1-2 Koto, Kamigori, Hyogo 678-1205, JAPAN
E-mail:yamaguti@lasti.u-hyogo.ac.jp
Tel/Fax: +81-791-58-0041
Reviewer 2 Report
Report: nanomaterials-1167983
In manuscript nanomaterials-1167983, Akinobu Yamaguchi, et al. reported the X-ray photoemission spectroscopy study of uniaxial magnetic anisotropy induced in a Ni layer deposited on a LiNbO3 substrate. Magnetic anisotropy is a foundation that many applications are based upon, efforts are thus constantly devoted to exploring (new) ways to control it. In this work, the authors get in touch with an interesting system, the heterojunction made out of ferromagnet/ferroelectric material. While the focus of this work is not on “how to control”, but the authors employ the XPS to rule out the formation of new chemical species at Ni/LiNbO3 heterojunction. That means whatever interface effect responsible for the uniaxial magnetic anisotropy reported in this particular system cannot be chemically originated. The data is convincing, I believe the community can be benefited from this work. However, I would like to bring forward a few questions and suggestions to the authors.
- This work means to explain the authors’ earlier findings reported in ref. 33-35. While it is nothing wrong with an article giving readers a clear introduction, but I would prefer to have it as concise as possible. I wonder is it a good idea to put Fig. 1 and a lengthy description on page 2 to re-iterate the findings that can be found in references. There is no new information that Fig. 1 can offer considering very similar results have been reported in ref. 34. Of course, this comment is more of a personal flavor.
- In Ln 139, it should be “Nb 3d” instead of “Nb 2p”.
- In Ln 140, there is no “240 eV” in Fig. 2(b). Maybe it is “204 eV”?
- 2(b)-(d) is somewhat confusing (at least to me).
- With Fig. 2 (b)-(d) describes the measurements taking place between etching time “930 s” to “1300 s”, the plots are marked differently as “interface -> substrate” in (b)&(d) and “Ni layer -> substrate”. What is the difference between “Ni layer” and “interface”? The term “interface” is not used in Fig. 5 & 6 when addressing similar scenarios.
- Fig. 2(b) shows three characteristic peaks, yet only the peak with a binding energy of 204 eV is marked as Nb2O5. I believe two of the three peaks in Fig. 2(b) stand for Nb 3d_3/2 and 3d_5/2, respectively. However, according to the Handbook of X-ray photoelectron spectroscopy, the characteristic peak stans for Nb2O5 should have higher binding energy (> 2 eV) than the 3d peaks. Can authors comment on their peak assignment and the subsequent discussion based on it?
- In Ln 163, how does the number “ 10 nm” come from if the authors only conduct measurement on specimens whose Ni thicknesses are 25 nm and 12.5 nm?
Author Response
Response to the Reviewer 2
We appreciate the Reviewer for instructive and helpful comments. We are very delighted to receive the Reviewer’s instructive criticisms. Following the Reviewer’s criticisms and comments, we have revised our manuscript. Below, we address our replies to all the issues raised together with the corrections in our revised manuscript:
[Reviewer’s criticisms (1)]
In manuscript nanomaterials-1167983, Akinobu Yamaguchi, et al. reported the X-ray photoemission spectroscopy study of uniaxial magnetic anisotropy induced in a Ni layer deposited on a LiNbO3 substrate. Magnetic anisotropy is a foundation that many applications are based upon, efforts are thus constantly devoted to exploring (new) ways to control it. In this work, the authors get in touch with an interesting system, the heterojunction made out of ferromagnet/ferroelectric material. While the focus of this work is not on “how to control”, but the authors employ the XPS to rule out the formation of new chemical species at Ni/LiNbO3 heterojunction. That means whatever interface effect responsible for the uniaxial magnetic anisotropy reported in this particular system cannot be chemically originated. The data is convincing, I believe the community can be benefited from this work. However, I would like to bring forward a few questions and suggestions to the authors.
This work means to explain the authors’ earlier findings reported in ref. 33-35. While it is nothing wrong with an article giving readers a clear introduction, but I would prefer to have it as concise as possible. I wonder is it a good idea to put Fig. 1 and a lengthy description on page 2 to re-iterate the findings that can be found in references. There is no new information that Fig. 1 can offer considering very similar results have been reported in ref. 34. Of course, this comment is more of a personal flavor.
[Our response (1)]
We appreciate the Reviewer’s severe and instructive criticisms. We half agree with the Reviewer’s opinion. Since both the single domain structure and the striped magnetic domain structure can be seen at the same time by the arrangement of the Ni wires in the same field of view, it is considered necessary to remind the reader that the alignment of Ni wires is important to control or form the domain structure. In addition, by showing the XAS spectra in Fig. 1(d), it also means that the Ni layer is not oxidized. Therefore, based on the above points, the explanation was simplified and shorted. In addition, the sentence that the Ni layer was not oxidized was described. As a result, we have revised these sentences in the 2nd paragraph of Introduction.
[Reviewer’s criticisms (2)]
1) In Ln 139, it should be “Nb 3d” instead of “Nb 2p”.
2) In Ln 140, there is no “240 eV” in Fig. 2(b). Maybe it is “204 eV”?
[Our response (2)]
We thank the Reviewer’s findings. Following the Reviewer’s criticisms, we have corrected them.
[Reviewer’s criticisms (3)]
2(b)-(d) is somewhat confusing (at least to me).
With Fig. 2 (b)-(d) describes the measurements taking place between etching time “930 s” to “1300 s”, the plots are marked differently as “interface -> substrate” in (b)&(d) and “Ni layer -> substrate”. What is the difference between “Ni layer” and “interface”? The term “interface” is not used in Fig. 5 & 6 when addressing similar scenarios.
Fig. 2(b) shows three characteristic peaks, yet only the peak with a binding energy of 204 eV is marked as Nb2O5. I believe two of the three peaks in Fig. 2(b) stand for Nb 3d_3/2 and 3d_5/2, respectively. However, according to the Handbook of X-ray photoelectron spectroscopy, the characteristic peak stans for Nb2O5 should have higher binding energy (> 2 eV) than the 3d peaks. Can authors comment on their peak assignment and the subsequent discussion based on it?
[Our response (3)]
We thank the Reviewer’s instructive criticisms. We absolutely agree with the Reviewer. We have checked the binding energy peaks of Nb, NbO, NbO2, Nb2O5, and LiNbO3. Following the Reviewer’s criticisms, we have calculated and reassigned the XPS peaks of Nb 3d. As a result, we have revised Figure 2 and the discussions from line 12 to line 35 in the 1st paragraph of 3. Results and discussion. And, we have added two references associated with the XPS spectra of Nb 3d.
[Reviewer’s criticisms (4)]
In Ln 163, how does the number “ 10 nm” come from if the authors only conduct measurement on specimens whose Ni thicknesses are 25 nm and 12.5 nm?
[Our response (4)]
We thank the Reviewer’s findings. We have revised it as the following: This result suggests that the electron state can only be modulated around the interface, as long as the Ni thickness is at least thicker than 12.5 nm.
We thank the Reviewer for his/her helpful comments. We have revised our manuscript accordingly. We would be most grateful if you could take a look at our revision and accept our responses and revision as appropriate.
Yours sincerely,
Akinobu Yamaguchi
---------------------------------------------------------------------
Akinobu Yamaguchi (Ph.D.)
Laboratory of Advanced Science and Technology for Industry, University of Hyogo 3-1-2 Koto, Kamigori, Hyogo 678-1205, JAPAN
E-mail:yamaguti@lasti.u-hyogo.ac.jp
Tel/Fax: +81-791-58-0041